# Presence of *trans*-Fatty Acids Containing Ingredients in Pre-Packaged Foods and the Availability of Reported *trans*-Fat Levels in Kenya and Nigeria

**DOI:** 10.3390/nu15030761

**Published:** 2023-02-02

**Authors:** Liping Huang, Adedayo E. Ojo, Judith Kimiywe, Alex Kibet, Boni M. Ale, Clementina E. Okoro, Jimmy Louie, Fraser Taylor, Mark D. Huffman, Dike B. Ojji, Jason H. Y. Wu, Matti Marklund

**Affiliations:** 1The George Institute for Global Health Australia, University of New South Wales, 1 King Street, Newtown, Sydney, NSW 2042, Australia; 2Cardiovascular Research Unit, University of Abuja Teaching Hospital, University of Abuja, Abuja 902101, Nigeria; 3Department of Epidemiology and Global Health, University Medical Centre, Utrecht University, 3508 Utrecht, The Netherlands; 4Center For Research Ethics and Safety, Kenyatta University, Nairobi P.O. Box 43844-00100, Kenya; 5Department of Nutrition and Dietetics, Kenya Medical Training College Karen Campus, Nairobi P.O. Box 24921, Kenya; 6Holo Healthcare, Nairobi P.O. Box 22003-00400, Kenya; 7Federal Capital Territory (FCT) Primary Health Care Board, Abuja 900001, Nigeria; 8School of Biological Sciences, The University of Hong Kong, Hong Kong, China; 9Department of Nursing and Allied Health, School of Health Sciences, Swinburne University of Technology, 1 John St., Hawthorn, VIC 3122, Australia; 10Department of Medicine and Global Health Center, Washington University in St. Louis, St. Louis, MO 63130, USA; 11Department of Preventive Medicine, Northwestern University, Chicago, IL 60611, USA; 12Department of Internal Medicine, Faculty of Clinical Sciences, University of Abuja, Abuja 900211, Nigeria; 13School of Population Health, University of New South Wales, Kensington, NSW 2052, Australia; 14Department of Epidemiology, Johns Hopkins Bloomberg School of Public Health, Baltimore, MD 21205, USA; 15Department of Public Health and Caring Sciences, Uppsala University, 75122 Uppsala, Sweden

**Keywords:** *trans*-fatty acids, packaged food, hydrogenation, cardiovascular disease

## Abstract

In most African countries, the prevalence of industrially produced *trans*-fatty acids (iTFA) in the food supply is unknown. We estimated the number and proportion of products containing specific (any hydrogenated edible oils) and non-specific (vegetable fat, margarine, and vegetable cream) ingredients potentially indicative of iTFAs among pre-packaged foods collected in Kenya and Nigeria. We also summarized the number and proportion of products that reported *trans*-fatty acids levels and the range of reported *trans*-fatty acids levels. In total, 99 out of 5668 (1.7%) products in Kenya and 310 out of 6316 (4.9%) products in Nigeria contained specific ingredients indicative of iTFAs. Bread and bakery products and confectioneries in both countries had the most foods that contained iTFAs-indicative ingredients. A total of 656 products (12%) in Kenya and 624 products (10%) in Nigeria contained non-specific ingredients that may indicate the presence of iTFAs. The reporting of levels of *trans-*fatty acids was low in both Kenya and Nigeria (11% versus 26%, respectively, p < 0.001). With the increasing burden of ischemic heart disease in Kenya and Nigeria, the rapid adoption of WHO best-practice policies and the mandatory declaration of *trans-*fatty acids are important for eliminating iTFAs.

## 1. Introduction

Cardiovascular disease is the leading cause of mortality and morbidity worldwide, and ischemic heart disease alone accounted for 16% of the world’s total mortality in 2019 [1]. A high intake of *trans*-fatty acids (>1% of total energy intake) [2] is a well-known risk factor for ischemic heart disease [3]. It is estimated that 500,000 deaths from ischemic heart disease each year globally are attributable to *trans*-fatty acid intake [2]. There are two main dietary sources of *trans*-fatty acids: (1) naturally occurring *trans*-fatty acids present in meat and dairy products and (2) industrially produced *trans*-fatty acids (iTFAs) through the partial hydrogenation of edible oils [4]. iTFAs do not add any nutritional value to the diet but increase low-density lipoprotein (LDL) cholesterol and decrease high-density lipoprotein (HDL) cholesterol [4], which in part increases the risk of coronary heart disease and other non-communicable diseases such as stroke.

The elimination of iTFAs was identified as a cost-effective intervention for the prevention of cardiovascular disease and could save 17.5 million lives globally over 25 years [5,6]. There are healthier options to replace iTFAs that do not impact the taste of food, making it feasible and acceptable to eliminate iTFAs from the food supply [7]. In 2018, World Health Organization (WHO) launched the REPLACE (Review, Promote, Legislate, Assess, Create, and Enforce) technical package to provide countries with a roadmap for eliminating iTFAs from national food supplies by 2023 [2]. Even before the launch of REPLACE in 2018, some high-income countries had already taken action to eliminate iTFAs. For example, Denmark in 2004 became the first country to put a limit on iTFAs (<2% of total fat) [5]. Since the launch of the WHO REPLACE package, significant progress has been made around the world [5].

In the African region, the burden of ischaemic heart disease has been increasing rapidly in the last two decades. For instance, ischemic heart disease is the number one killer in Nigeria [8] and the second leading cause of death in Kenya [9] among non-communicable diseases. In the meantime, it is well recognized that the world has undergone a nutritional transition towards more processed and pre-packaged foods for convenience and affordability, and this transition pace is the fastest in the Sub-Saharan areas, where multiple social factors such as urbanization and disposable income growth are driving the change [10]. Consequently, the increasing consumption of processed and pre-packaged foods, which are high in fat, sodium, and calories [11,12,13], is leading to a higher risk of diet-related diseases in the region. For instance, there was a 27% increase in Nigeria [8] and a 33% increase in Kenya [9] in death and disability combined attributable to dietary risks from 2009 to 2019, making dietary risks one of the top 10 contributing factors towards death and disability.

With regards to iTFAs, although there was national policy commitment to eliminate iTFAs from several African countries, up to December 2022, only South Africa in the African region had implemented a WHO best-practice policy (i.e., legislative or regulatory measures that limit iTFAs in foods in all settings and are in line with the recommended approach) [14]. Between 2010 (i.e., the year before the policy was implemented) and 2019, the number of deaths from ischemic heart disease in South Africa remained stable (with a 21% decrease in age-standardized mortality rate) [15]. During the same period, the number of deaths from ischemic heart disease in two other major countries in the Sub-Saharan region, Nigeria and Kenya, increased by 23% and 33%, respectively [15]. While both Kenya and Nigeria have committed to policies against iTFAs, these have so far not been implemented [16,17]. Furthermore, the prevalence of iTFAs in the two countries’ food supply is unknown due to the lack of prior investigation.

The presence of *trans-*fatty acids in packaged foods can be communicated through nutrition labels on food packages. Nutrition labeling is a way to provide consumers with the nutritional information of food products and thus assist consumers in making informed choices. As part of nutrition labeling, nutrient declaration shows the amount of different nutrients contained in per 100 g or per serving food product and is traditionally displayed at the back of the package. Globally, over 80 countries require mandatory declarations of various nutrients such as energy, protein, sodium, fat, sugar, and carbohydrates [18]. However, in many African countries [18], including Kenya and Nigeria [19,20], mandatory nutrient declaration on food packages is only required when the food makes health or nutrient claims. Although it is challenging to quantify the level of *trans*-fatty acids in food without mandatory nutrition declarations on packaged products, it is possible to identify food products potentially containing iTFAs based on the ingredient list (e.g., partially hydrogenated vegetable oils) [21]. In this study, we used the ingredient list of packaged food to assess the prevalence of iTFA-indicative ingredients in Kenya and Nigeria, two major countries in Eastern Africa and Western Africa, respectively, together home to roughly every fifth person in Africa [22]. We also estimated the proportion of products with reported levels of *trans*-fatty acids and recorded the ranges of reported values.

## 2. Materials and Methods

### 2.1. Data Sources

The FoodSwitch project at the George Institute for Global Health is a large international project that collects detailed product and nutritional information about packaged foods in 17 countries [23]. Countries that participate in the project follow standard data collection and processing protocols which have been previously described [24,25]. In each participating country, data were collected from large retailers at a one-time point or annually by trained staff using a smartphone data collection application. In each selected store, data on all food available for sale were collected. Images of packaged foods were taken using a smartphone data collection application and sent to a bespoke content management system where data were extracted from the images and stored using a quality control system. According to CODEX ALIMENTARIUS International Food Standards, it is mandatory to include a list of ingredients in the food package as part of the general food labelling regulation for packaged food except for single-ingredient food [26]. Therefore, ingredient information of products was also entered for products with clear images. All food products were categorized according to a hierarchical food categorization system that contains 18 major food groups (alcoholic beverages; bread and bakery products; cereal and grain products; confectioneries; convenience foods; dairy; edible oils and oil emulsions; eggs, seafood and seafood products; fruits, vegetables, nuts, and legumes; meat and meat products; non-alcoholic beverages; sauces, dressings, spreads, and dips; snack foods; special foods; sugars, honey, and related products; vitamins and supplements; unable to be categorized) and over 1000 finer sub-categories, as defined by the Global Food and Nutrition Monitoring group [25]. For the purpose of this analysis, eggs and alcoholic beverages were not included, as eggs do not have ingredient information, and alcoholic beverages are not in the scope of this research, leaving 16 major food groups for analysis.

Data used in this analysis were collected in Kenya and Nigeria as part of the FoodSwitch project. For Kenya, data were collected in 2019 from five major supermarket chains in Nairobi, Kenya (Tuskys, Chandarana, Kassmatt, Stanmatt, and Kamindi Selfridges) [19]. For Nigeria, data were collected from major supermarkets (Shopright, Grandsqure supermarket and Stores Limited, Market Square, Prince Ebeano Justrite) across three states (Federal Capital Territory, Kano, Ogun) from November 2020 to March 2021. The selection of stores for data collection was determined by local collaborators in each country based on their research and knowledge of the size (largest in size) of the stores and the frequency of customer visits (most frequently visited).

### 2.2. Analysis

Some products were exempted from displaying ingredient information, and some did not have clear images to allow data entry; only products with ingredient information were included in this analysis. Following the same method of previous research [21,27], the ingredient list of each product was searched using specific terms (i.e., partially hydrogenated fat, hydrogenated vegetable oil, and hydrogenated) that indicate ingredients that are sources of iTFAs. In addition, non-specific terms (i.e., vegetable fat, margarine, and vegetable cream), which may or may not indicate iTFA-containing ingredients, were also searched [27]. Manual checks were also conducted after an automatic search by programming to correct misclassification due to variations and misspellings of ingredient information. The number and proportion of foods containing specific and non-specific iTFA-indicating ingredients were calculated across all products and for each of the 16 major food groups. The availability and level of reported *trans*-fatty acids on the nutritional information panel (NIP) among all products and among products that contained specific iTFA-indicating ingredients were also examined. Pearson’s chi-squared test was conducted to see if there were differences between Kenya and Nigeria, in terms of the prevalence of iTFA-indicating products and the prevalence of *trans*-fatty acid information on NIP. Two-sided *p* values < 0.05 were considered statistically significant without adjustment for multiple testing. Analyses were done using R version 4.1.0 and RStudio 2 July 2022 Build 576 [28]. Packages “plotrix” [29], “gtsummary” [30], and “janitor” [31] were used for summarizing the data.

## 3. Results

In total, data were collected from 6711 pre-packaged products in Kenya in 2019 and from 7039 pre-packaged products in Nigeria between November 2020 and March 2021. Of these, 5668 and 6316 non-alcoholic products with ingredient information were included in this analysis for Kenya and Nigeria, respectively.

### 3.1. Presence of Specific Ingredients Indicative of Industrially Produced trans-Fatty Acids in Kenya and Nigeria

A total of 99 out of 5668 products (1.7%) in Kenya contained specific ingredients that are indicative of iTFAs. This proportion was statistically higher (*p* < 0.001) in Nigeria, with a total of 310 out of 6316 products (4.9%) that contained specific ingredients indicative of iTFAs. In Kenya, the top four food groups that contained such ingredients were bread and bakery products (*n* = 40, 5.3% of products); confectionery (*n* = 13, 6.7%); sauces, dressings, spreads, and dips (*n* = 13, 2.6%) and non-alcoholic beverages (*n* = 13, 1.6%). In Nigeria, the top four food groups were confectioneries (*n* = 98, 16.9% of products); bread and bakery products (*n* = 73, 6.3%); non-alcoholic beverages (*n* = 44, 5.4%); and snack foods (*n* = 25, 7%) (Table 1). Over 50% of the products with specific ingredients indicative of iTFAs in Kenya were produced by 10 manufacturers, and about 40% of products with specific ingredients indicative of iTFAs in Nigeria were produced by 10 manufacturers. 

### 3.2. Presence of Ingredients Potentially Indicative of Industrially Produced trans-Fatty Acids in Kenya and Nigeria

A total of 656 out of 5668 products (11.6%) in Kenya and a slightly smaller number and proportion (624 out of 6316 products (9.9%), *p* = 0.003) in Nigeria contained non-specific ingredients that may indicate the presence of iTFAs. In Kenya, the top three food groups with the most products were bread and bakery products (*n* = 432, 57.3% of products); dairy (*n* = 71, 7.6%); and confectioneries (*n* = 64, 32.8%). In Nigeria, the top three food groups were bread and bakery products (*n* = 304, 26.1% of products); confectioneries (n = 162, 27.9%); and fruits, vegetables, nuts, and legumes (*n* = 55, 5.9%) (Table 1). The number and proportion of products containing each ingredient are presented in Appendix Table A1.

### 3.3. Availability and Levels of Reported trans-Fatty Acids in Kenya and Nigeria

Of all products included in this analysis, 584 (10%) products in Kenya reported the level of *trans*-fatty acids on the NIP. A larger number and proportion (1626 (26%), *p* < 0.001) of products in Nigeria reported the level of *trans*-fatty acids on the NIP. There was a large variation in the proportion of food reporting the levels of *trans*-fatty acids across different food groups. The food categories most commonly reporting *trans*-fatty acid levels were edible oils and oil emulsions in Kenya (28%) and snack foods in Nigeria (37%) (Figure 1). Among products with reported *trans*-fatty acids levels, the level of *trans*-fatty acids ranged from 0 to 2 g per 100 g product in Kenya and ranged from 0 to 6.5 g per 100 g product in Nigeria. The average proportion of *trans*-fatty acids out of total fat was 0.58% in Kenya and 0.28% in Nigeria. The number of products with reported *trans*-fatty acids exceeding 2% of total fat was 40 (6.8%) products in Kenya and 48 (3%) products in Nigeria (Table 2).

Less than half of products with specific ingredients indicative of iTFAs reported *trans*-fatty acids levels (40% in Kenya and 41% in Nigeria, *p* > 0.9), and of these, a few products (8 (20%) in Kenya, and 9 (7.1%) in Nigeria) exceeded 2% of total fat.

## 4. Discussion

This analysis found that overall, less than 2% of pre-packaged food products sold in major supermarket chains in Kenya and nearly 5% of pre-packaged food products sold in major supermarkets in Nigeria contained specific ingredients indicative of iTFAs. In both countries, products containing such ingredients were clustered in a few major food groups, particularly bread and bakery products and confectionery (e.g., 17% of confectioneries in Nigeria included specific iTFA-containing ingredients). In addition, the proportion of products declaring the level of *trans*-fatty acids was low, and there was no consistent reporting across different food groups, which makes benchmarking the amount of *trans*-fatty acids contained in the food challenging. The low reporting rate of *trans*-fatty acids levels among foods containing specific ingredients indicative of iTFAs is particularly worrisome, with only 4 in 10 of such products displaying information on *trans*-fatty acids content in both Kenya and Nigeria.

Although the proportion of products containing specific ingredients indicative of iTFAs is small in both Kenya and Nigeria, the impact on public health would still be significant if the products were consumed in large quantities. For instance, bread and bakery products are major foods in many countries, including African countries. The per capita consumption of bread in Kenya was 19 kg in 2015 [32]. The market for bread and bakery products was expected to grow by 6.9% in African countries from 2016 to 2025, while the Kenyan market is expected to grow by 11% annually for a period of 20 years [33]. Thus, it is possible that iTFAs intake (and the health consequences thereof) may increase with a greater intake of bread and bakery products. Our analysis showed that most products containing specific iTFA-containing ingredients are clustered in a few food groups, and a large proportion of them was produced by a limited number of manufacturers. If analysis of nationally representative samples of packaged foods shows similar patterns, removal of iTFAs from the food supply may be feasible with targeted efforts. However, a mandatory declaration of *trans*-fatty acids will be important to monitor and revise the targeted food groups for efforts over time.

In response to WHO’s global target to eliminate iTFAs, countries are moving fast in developing relevant policies. Currently, mandatory iTFA limits or bans on partially hydrogenated oils are in effect in 57 countries, protecting 3.2 billion people [5]. As eliminating iTFAs is relatively feasible, acceptable, and affordable due to the limited number of products with iTFAs and requires relatively little resource, even some low- and lower-middle-income countries have passed policies to regulate iTFAs. For example, India passed a policy effective as of January 2022, and a best-practice policy in the Philippines is expected to take effect in July 2023 [5]. Although both Kenya [17] and Nigeria [16] have outlined policies to limit iTFAs in foods, alignment with WHO best-practice policy is yet to be adopted [14]. 

Both Kenya and Nigeria are experiencing an increasing burden of ischaemic heart disease. According to the Global Burden of Disease Study, deaths attributable to ischaemic heart disease increased by 37% in Kenya [9] and increased by 25% in Nigeria [8] from 2009 to 2019. The elimination of iTFAs from the food supply could reduce both the burden of ischaemic heart disease and deaths from other causes where prevalent ischaemic heart disease may increase mortality risks (e.g., people with underlying health issues such as ischaemic heart disease were at higher risk of dying due to complications of COVID-19) [34]. Prior modelling work done for Australia suggested that iTFAs elimination could be a cost-effective, or even cost-saving, strategy to reduce the ischaemic heart disease burden, preventing around 2300 deaths attributable to the disease over the first ten years [35]. Dietary patterns and other contexts may be different in Kenya and Nigeria compared to Australia and hence will require specific modelling work to be done for the two countries to properly assess the cost-effectiveness of iTFAs elimination. Nonetheless, following WHO’s recommended best-practice policy, which includes: (1) a mandatory national limit of 2 g of iTFAs per 100 g of total fat in all foods and (2) a mandatory national ban on the production or use of partially hydrogenated oil [5], may have a meaningful impact on the countries’ rising burden of ischaemic heart disease [36], particularly when there is a rapid dietary transition of moving from traditional fresh food to processed and pre-packaged foods. In addition, mandatory declaration of *trans*-fatty acids on food packages will be important for quantifying the level of *trans*-fatty acids contained in the food supply, benchmarking progress towards eliminating iTFAs, and helping consumers make healthier choices at the point of purchase.

This paper benefits from the standard data collection of packaged foods in Kenya and Nigeria following the same protocol and provides useful insights into the presence of iTFA-indicative ingredients and the presence of *trans*-fatty acids information on food packages. Although the sample size included for both countries are relatively large, the data may not fully capture the variability in the use of iTFAs ingredients and levels of *trans*-fatty acids in the food supplies in these countries. On the other hand, the data were collected from major retailers where a large proportion of products are sold and most of which would be subsequently consumed. Nonetheless, future analysis incorporating sales data of different products would have added value to understanding the purchase pattern and real impact of the products containing iTFAs. In addition, the paper only focused on pre-packaged foods and did not include freshly baked goods and may therefore underestimate the presence of iTFAs in the overall food supply [37]. Also, the analysis of the non-specific iTFA-indicating ingredients should be interpreted cautiously and should ideally be validated by the use of chemical analysis of *trans*-fatty acids levels in such products. We relied on the accuracy and validity of the nutritional information available on the NIP of the food products captured in the database for both Kenya and Nigeria. The reported levels of *trans*-fatty acids in this paper should be used with caution as only a small proportion of products reported the levels of *trans*-fatty acids on the NIP, and this reporting may be selective depending on the true levels of *trans*-fatty acids contained in the food. We reported the proportion of products with *trans-*fatty acids content exceeding 2% of total fats, and it is possible that the iTFAs content of such products may be less than 2% of total fats, especially in foods containing significant amounts of naturally occurring *trans*-fatty acids (e.g., some dairy foods).

## 5. Conclusions

Although only a small proportion of all packaged food products in Kenya and Nigeria contained specific ingredients indicative of iTFAs, the potential impact on public health may be significant given the increasing market share and greater demand for packaged food. Reporting of *trans*-fatty acids levels was overall scarce, even among products containing specific iTFA-indicative ingredients, suggesting a need for more efforts in Africa towards mandatory labelling of *trans*-fatty acid content of packaged foods. The rapid adoption of WHO best-practice policy for both countries to eliminate iTFAs may be important given the rising burden of ischaemic heart disease in both countries. Governmental efforts will be crucial to achieving iTFAs elimination, as previous research showed that only full compliance with relevant regulations showed success [38,39,40]. In addition, it will be critical to mandate the declaration of *trans*-fatty acids on the NIP in Kenya and Nigeria, as this not only allows a better understanding of the presence of iTFAs and facilitates the monitoring of the progress towards iTFAs elimination but also has been shown to result in reductions of *trans*-fatty acid in blood levels [41,42,43]. Mandatory declaration will also help consumers make their food choices.

## Figures and Tables

**Figure 1 nutrients-15-00761-f001:**
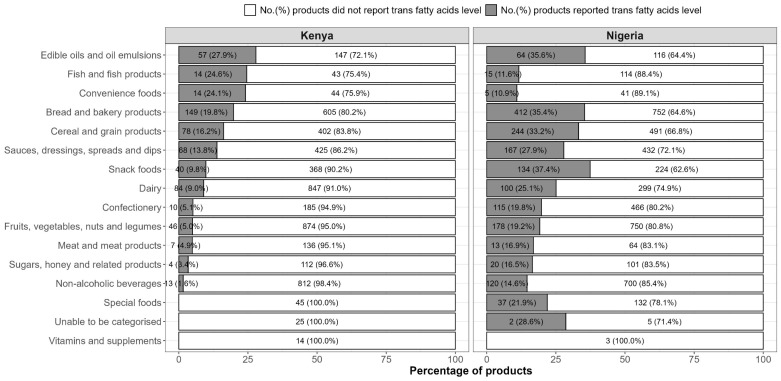
Number and percentage of products reported and did not report trans-fatty acids level by major food groups in Kenya and Nigeria.

**Table 1 nutrients-15-00761-t001:** Presence of iTFA-indicative ingredients in pre-packaged foods in Kenya and Nigeria.

Food Group	Kenya			Nigeria		
	Total *n*	No. (%) of Products Contains Any Specific Terms Indicating iTFAs	No. (%) of Products Contains Any Non-Specific Terms Indicating iTFAs	Total *n*	No. (%) of Products Contains Any Specific Terms Indicating iTFAs	No. (%) of Products Contains Any Non-Specific Terms iTFAs
Bread and bakery products	754	40 (5.3)	432 (57.3)	1164	73 (6.3)	304 (26.1)
Cereal and grain products	480	2 (0.4)	36 (7.5)	735	7 (1.0)	3 (0.4)
Confectionery	195	13 (6.7)	64 (32.8)	581	98 (16.9)	162 (27.9)
Convenience foods	58	3 (5.2)	4 (6.9)	46	1 (2.2)	0 (0.0)
Dairy	931	9 (1.0)	71 (7.6)	399	13 (3.3)	46 (11.5)
Edible oils and oil emulsions	204	0 (0.0)	5 (2.5)	180	8 (4.4)	7 (3.9)
Seafood and seafood products	57	0 (0.0)	4 (7.0)	129	0 (0.0)	0 (0.0)
Fruits, vegetables, nuts and legumes	920	4 (0.4)	0 (0.0)	928	19 (2.0)	55 (5.9)
Meat and meat products	143	0 (0.0)	3 (2.1)	77	0 (0.0)	3 (3.9)
Non-alcoholic beverages	825	13 (1.6)	7 (0.8)	820	44 (5.4)	23 (2.8)
Sauces, dressings, spreads and dips	493	13 (2.6)	15 (3.0)	599	19 (3.2)	9 (1.5)
Snack foods	408	2 (0.5)	11 (2.7)	358	25 (7.0)	10 (2.8)
Special foods	45	0 (0.0)	0 (0.0)	169	0 (0.0)	2 (1.2)
Sugars, honey and related products	116	0 (0.0)	0 (0.0)	121	2 (1.7)	0 (0.0)
Vitamins and supplements	14	0 (0.0)	1 (7.1)	3	0 (0.0)	0 (0.0)
Unable to be categorized	25	0 (0.0)	3 (12.0)	7	1 (14.3)	0 (0.0)
Total	5668	99 (1.7)	656 (11.6)	6316	310 (4.9)	624 (9.9)

Note: Specific terms include “Partially Hydrogenated Fat”, “Hydrogenated Vegetable Oil” and “Hydrogenated” and exclude “Fully Hydrogenated”. Non-specific terms include “Vegetable Fat”, “Margarine”, and “Vegetable Cream”. Special foods are foods for special dietary use, including meal replacements, sports foods, infant and baby food, etc. iTFAs, industrially produced trans-fatty acids.

**Table 2 nutrients-15-00761-t002:** Levels of reported trans fat in Kenya and Nigeria for different food groups.

Food Group	Kenya			Nigeria		
	No. (%) Products with *trans*-Fatty Acids > 2% of Total Fat	Median (Range) of Reported *trans*-Fatty Acids Levels (g/100 g)	Mean (sd) Percentage (%) *trans*-Fatty Acids of Total Fat	No. (%) Products with *trans*-Fatty Acids > 2% of Total Fat	Median (Range) of Reported *trans*-Fatty Acids Levels (g/100 g)	Mean (sd) Percentage (%) *trans*-Fatty Acids of Total Fat
Bread and bakery products	9 (1.2)	0.00 (0.00, 1.00)	0.25 (0.78)	6 (0.5)	0.00 (0.00, 6.5)	0.19 (1.60)
Cereal and grain products	1 (0.2)	0.00 (0.00, 0.30)	0.22 (0.71)	2 (0.3)	0.00 (0.00, 1.00)	0.31 (4.27)
Confectionery	0 (0.0)	0.00 (0.00, 0.30)	0.25 (0.40)	5 (0.9)	0.00 (0.00, 6.30)	0.50 (1.97)
Convenience foods	5 (8.6)	0.01 (0.00, 0.85)	2.10 (3.26)	0 (0.0)	0.00 (0.00, 0.00)	0.00 (0.00)
Dairy	18 (1.9)	0.00 (0.00, 1.68)	2.14 (10.93)	17 (4.3)	0.00 (0.00, 2.04)	0.78 (1.58)
Edible oils and oil emulsions	0 (0.0)	0.10 (0.00, 0.70)	0.30 (0.31)	6 (3.3)	0.00 (0.00, 3.20)	0.61 (1.10)
Fish and fish products	3 (5.3)	0.00 (0.00, 0.20)	1.55 (3.87)	1 (0.8)	0.00 (0.00, 0.10)	0.32 (0.96)
Fruits, vegetables, nuts and legumes	2 (0.2)	0.00 (0.00, 0.50)	0.18 (0.56)	4 (0.4)	0.00 (0.00, 3.60)	0.09 (0.69)
Meat and meat products	0 (0.0)	0.00 (0.00, 0.00)	0.00 (0.00)	1 (1.3)	0.00 (0.00, 0.30)	0.46 (1.66)
Non-alcoholic beverages	0 (0.0)	0.00 (0.00, 0.00)	0.00 (0.00)	0 (0.0)	0.00 (0.00, 0.05)	0.03 (0.16)
Sauces, dressings, spreads and dips	2 (0.4)	0.00 (0.00, 2.00)	0.31 (1.70)	1 (0.2)	0.00 (0.00, 0.10)	0.30 (3.87)
Snack foods	0 (0.0)	0.00 (0.00, 0.10)	0.05 (0.32)	5 (1.4)	0.00 (0.00, 2.00)	0.34 (1.82)
Special foods	-	-	-	0 (0.0)	0.00 (0.00, 0.03)	0.01 (0.04)
Sugars, honey and related products	0 (0.0)	0.00 (0.00, 0.00)	0.00 (0.00)	0 (0.0)	0.00 (0.00, 0.00)	0.00 (0.00)
Vitamins and supplements	-	-	-	-	-	-
Unable to be categorized	-	-	-	0 (0.0)	0.00 (0.00, 0.00)	0.00 (0.00)
Overall	40 (0.7)	0.00 (0.00, 2.00)	0.58 (4.33)	48 (0.8)	0.00 (0.00, 6.5)	0.28 (2.40)

## Data Availability

The data supported this research was collected as part of the FoodSwitch project. Data may be shared within reasonable requests. For any queries, please contact Fraser Taylor at ftaylor@georgeinstitute.org.au.

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
