# Peer review of "Presence of trans-Fatty Acids Containing Ingredients in Pre-Packaged Foods and the Availability of Reported trans-Fat Levels in Kenya and Nigeria"

_nutrients, 2023, doi:10.3390/nu15030761_

Round 1

Reviewer 1 Report

This paper is making an important contribution to the literature as there is scarce data on iTFAs in African countries. Below, please find comments and suggestions to increase the paper's value. 

Intro, line 52: I would use more standard language to describe the intervention. Rather than say 'best buy', maybe say that it has beenidentified to be a 'cost-effective' or "cost-saving" intervention? 

Intro, line 64: You say that South Africa is the only African country to have adopted the WHO best practice policy. It would be nice if you plotted the cardiovascular disease mortality (CVDM) rates for Africa (overall), South Africa, Nigeria, and Kenya in the Introduction. Doing so would allow you to discuss the potential health benefits of adopting the same practice in Nigeria and Kenya versus South Africa. For example, if the CVDM rates are higher in Kenya than in South Africa, there is a strong argument for Kenya to quickly follow in the footsteps of South Africa. The proposed figure would be a nice complement to the other data you provided in the Introduction regarding how Kenya and Nigeria combined represent about 20% of Africa's population. 

Data sources, lines 92-96: Do you have any information on what share of the retail food market are represented by the supermarket chains contributing data to the FoodSwitch project? I think this information would go a long way in helping readers understand how well the data capture the extent of the iTFA problem in Kenya and Nigeria. I would also explicitly state that the data you use are not nationally representative, though cover an important part of the retail food market. 

Discussion, lines 189-193: The data you use are not representative of all grocery stores in Kenya/Nigeria, so I would recommend adding some caveats here. If analysis of more nationally representative data show patterns across food groups that mirror those in your paper, then a targeted iTFA removal policy would perhaps be effective. However, it would also be important to know more about the trans fat content of foods in all packaged foods offered for sale -- so as you note on lines 218 -- any iTFA removal policy should be accompanied by mandatory declaration of TFA contents on all packaged foods to help monitor their presence in the entire food supply. Doing so would allow food authorities to revise the food groups targeted as needed if a tailored iTFA removal policy is ultimately adopted. 

Discussion, lines 227-228: I would suggest removing "and subsequently consumed". You cannot know how much of the purchased food people actually consume. The correspondence between purchases and consumption is not always one-to-one as some food is wasted and some food is shared with others. Both of these actually reduce the dietary risks of the total amount of purchased food bought by the individual.

Conclusions, line 248-249: I would soften the statement about there being a "strong" need for African countries to eliminate iTFAs. I would suggest saying that there is a critical need to know the iTFA content in all packaged foods (mandatory TFA declarations) to better understand the extent of the TFA problem in Kenya/Nigeria. You might also cite published research that shows that in other countries such as the USA, mandatory labeling of (overall) TFA content was followed by reductions in the presence of iTFA in the blood levels of people living in the USA. (For example, see https://doi.org/10.1001/jama.2012.112, https://doi.org/10.2105/AJPH.2016.303524, and https://doi.org/10.1017/S1368980019003367). I would then say that adopting WHO's best practice policy to remove iTFAs would reduce CVDM risks and the TFA declarations on packaged foods can be used for surveillance. Therefore, the mandatory declarations can achieve three objectives -- encourage healthier product reformulations by food producers, arm consumers with more information to make food choices, and allow food authorities to track the presence of iTFAs in the food supply over time. You might also cite work that shows that governmental efforts to reduce iTFAs in the food supply have been shown to work in some jurisdictions like NYC (https://doi.org/10.1016/j.jhealeco.2015.09.005) and Denmark (https://doi.org/10.1016/j.amepre.2015.06.018), but not in others like Austria where there wasn't full compliance with TFA regulations (https://doi.org/10.1093/eurpub/cky147)

Author Response

We thank the reviewer's time and effort in reviewing our manuscript. The responses to each comment are enclosed in the attachment. 

Reviewer 2 Report

This is a well-crafted study that will make a useful contribution to the literature. The manuscript would be strengthened by the inclusion of an overview of the current context in Nigeria and Kenya, considering present labelling regulations, public health messaging, and food consumption patterns as these relate to iTFA.

L43-44: How important is IHD as a cause of death in Africa – and Kenya and Nigeria in particular? I can see that this information appears in the discussion, but I suggest at least some of it be moved to the introduction. It is central to the rationale for this study.

L93-94: If available, please include information about market share.

L92-96: A major consideration in a study like this is the representativeness of the stores sampled. Readers need to be provided with more information about these samples in order to gauge what is being reported. What does ‘major’ mean and how was it determined? Were the chains sampled in Nairobi operating all over the country? What were the supermarkets or supermarket chains from which data were collected in Nigeria and how were they identified?

L98: What products in these countries would have ingredient lists on the label? Please briefly describe and/or reference the regulations governing labelling. To put the data collection in perspective, readers need to understand the regulatory context for this study. It would also be helpful to know what if any public health messaging is present in these countries with respect to trans fats. Are consumers being advised to avoid foods with iTFA?

L101-103: This is not a sentence.

L107-109: What are the regulations concerning the presentation of NIP and declaration of trans fatty acid levels on this panel in these countries?  

L117-120: Why did so many pre-packaged products not have ingredient information?

L178-181: Were products containing specific ingredients indicative of iTFA more or less likely to display information about TFA on their NIPs than products not containing such ingredients? It would be interesting to see this comparison as the information in table 2 suggests that most of the products with labelled values had zero TFA. This is consistent with a much larger body of research indicating that voluntary nutrition labelling is primarily a marketing tool for industry.

L182-190: Because the trans fat content of products consumed on a daily basis is of most concern, it would be helpful to see information about how they relate to usual dietary intakes in the country, if such data are available.

L191-192: How do you know that most of these products are produced by major manufacturers?

L196-198: If the products impacted are staple foods for a large swath of the population, then I think the measures required to eliminate iTFA are unlikely to be as simple as the authors imply. Insofar as product reformulation increases food prices for lower income households, iTFA elimination may not be inconsequential. Presumably this issue would be factored into the cost-effectiveness analysis discussed in the next paragraph.

L210-212: How relevant the results of modelling work in Australia are to Nigeria or Kenya surely depends on the similarities and differences in these populations - their food consumption patterns, regulatory contexts, and other public health measures in place to discourage the use of iTFA-containing products? Readers really need more information about the current context in Nigeria and Kenya.

L222-229: How comparable were the stores sampled in the two countries? How representative are the products sold in these stores of the products sold elsewhere in these countries?

L222-242: I think it is also important to comment on the lack of information about market share and price. The current presentation of data treats all products as equal, but the trans fat concentrations of products that are consumed regularly and in substantial amounts matter much more than products only occasionally consumed. For example, if the trans fat-containing bakery products are the cheapest ones and therefore the ones most widely consumed, the fact that they comprise only a small fraction of the total universe of bakery products sampled is unimportant.

L245-247: Is the diversity observed worth including as a conclusion? Would anyone have expected otherwise?

L248-254: The current regulatory context in each country needs to be described earlier on, so that readers can understand what changes you are recommending and why.

Author Response

We thank the reviewer's time and effort in reviewing our manuscript and providing feedback. We have enclosed the responses to each comment in the attachment. 
